# Effects of Hydrogen Plasma Treatment on the Electrical Behavior of Solution-Processed ZnO Thin Films

**DOI:** 10.3390/ma17112673

**Published:** 2024-06-01

**Authors:** Ji-In Park, Hyun Uk Lee, Christopher Pearson, Michael C. Petty, Yesul Jeong

**Affiliations:** 1Research Center for Materials Analysis, Korea Basic Science Institute (KBSI), Daejeon 34133, Republic of Korea; jipark@kbsi.re.kr (J.-I.P.); leeho@kbsi.re.kr (H.U.L.); 2School of Engineering and Computing Sciences and Centre for Molecular and Nanoscale Electronics, Durham University, South Road, Durham DH1 3LE, UK; christopher.pearson@durham.ac.uk; 3Busan Center, Korea Basic Science Institute (KBSI), Busan 46742, Republic of Korea

**Keywords:** zinc oxide, solution process, plasma treatment, effect of measurement condition, in-plane electrical conductivity

## Abstract

In this study, the effect of atmospheric hydrogen plasma treatment on the in-plane conductivity of solution-processed zinc oxide (ZnO) in various environments is reported. The hydrogen-plasma-treated and untreated ZnO films exhibited ohmic behavior with room-temperature in-plane conductivity in a vacuum. When the untreated ZnO film was exposed to a dry oxygen environment, the conductivity rapidly decreased, and an oscillating current was observed. In certain cases, the thin film reversibly ‘switched’ between the high- and low-conductivity states. In contrast, the conductivity of the hydrogen-plasma-treated ZnO film remained nearly constant under different ambient conditions. We infer that hydrogen acts as a shallow donor, increasing the carrier concentration and generating oxygen vacancies by eliminating the surface contamination layer. Hence, atmospheric hydrogen plasma treatment could play a crucial role in stabilizing the conductivity of ZnO films.

## 1. Introduction

Solution-processed zinc oxide (ZnO) is emerging as a critical semiconductor material with potential applications in various fields, such as field-effect transistors (FETs), gas sensors, and memories in flexible large-area electronics. These ZnO nanostructures have been synthesized using various physical and chemical manufacturing techniques, including carbon-thermal transport growth, electron beam evaporation, vapor–liquid–solid (VLS), and hydrothermal processes. The carbon-thermal transport growth allows easy control of layer thickness; however, it is complex and expensive [1]. The VLS process, which is the simplest and most efficient suffers from long reaction times and low reproducibility [2]. Electron beam evaporation methods using high-vacuum, high-voltage systems have fast deposition rates; however, they involve high energy costs [3]. Hydrothermal methods require long reaction times and harsh conditions such as high temperature and pressure, which limits their applications in next-generation research fields [4]. Recently, a solution coating technology that uniformly produces films of various thicknesses with a facile, low-energy-cost process has garnered considerable attention [5,6].

For most electrical devices, the material should be present as a thin film, highly sensitive to ambient conditions [7]. The absorbed molecules from the environment, especially oxygen molecules, decrease the electrical properties owing to surface energy band bending [8]. Moreover, film thickness is a crucial factor in determining the electrical properties and environmental sensitivity. In the case of a thick film, the thickness is sufficient to create a conductive path in the bulk region of the film because the film comprises uniform, large-sized, and dense grains. Further, the bulk defect is the dominant factor determining the electrical properties, as the surrounding environment does not considerably affect thick films. However, films with thicknesses <20 nm are too thin to form a continuous conductive path through the bulk region of the film. Two dominant factors determine the conductivity of these films: surface and bulk defects. Several studies have reported that reactions with surface defects and exposure to ambient air lead to conductivity degradation [9,10,11,12,13,14,15].

Recently, several techniques for H_2_-doping of ZnO thin films, including annealing in an H_2_ environment [16], exposure to H_2_ plasma [17,18], and H_2_-burial during deposition [19], have been used to improve the electrical conductivity of thin films. Hydrogen acts as an excellent alternative dopant in ZnO for several reasons. Similar to group III elements (Al, Ga, and In), which act as shallow donors in ZnO, a small quantity of hydrogen doping improves the crystallinity, increases the band gap, and decreases the electrical resistance of ZnO films [20]. Ding et al. reported a mobility (μ) of 46 cm^2^/Vs by exposing H_2_ plasma to 350 nm thick ZnO using plasma-enhanced chemical vapor deposition (PECVD) [18]. Macco et al. prepared ZnO:H films with μ = 47 cm^2^/Vs and ρ = 1.8 mΩ cm using interleaved standard atomic layer deposition (ALD) of ZnO cycles comprising diethylzinc (DEZ, Zn(C_2_H_5_)_2_) and H_2_O via plasma exposure [21]. However, this H_2_ plasma treatment method using the vacuum process is time-consuming and requires harsh conditions, such as high temperature and pressure [22]. Recently, majority of the studies on conductivity mechanisms have focused on vacuum-processed ZnO films; hence, investigating the effects of exposing solution-processed ZnO films to different environments is crucial.

In this study, we investigate the effect of hydrogen plasma treatment on the in-plane electrical conductivity when solution-processed ZnO thin films are exposed to various environments. We believe that our work will contribute remarkably to the existing literature because it involves a systematic and comprehensive study addressing the environmental sensitivity of the films and thus develops a better understanding of the electrical conductivity of solution-processed ZnO films.

## 2. Materials and Methods

The zinc oxide solution was prepared using a previously reported method [23,24,25,26,27,28]. Briefly, zinc hydroxide was synthesized from zinc nitrate, Zn(NO_3_)_2_ (Duksan, Ansan-si, Republic of Korea, 95%, used without further purification). The synthesized Zn(OH)_2_ powder was solubilized by dissolving it in approximately 5% ammonia water. The solution was spin-coated (3000 rpm, 30 s) to a thickness of approximately 20 nm. Subsequently, it was annealed at 140 °C for 30 min in air on a hotplate. The films were exposed to an atmospheric cold plasma system (AP100, FemtoScience, Hwaseong-si, Republic of Korea) for 10, 15, or 30 min at an RF power of 100 W under hydrogen gas. The in-plane and DC electrical conductivities of the solution-processed ZnO films were monitored using a Keithley 2400 SourceMeter (KEITHLEY, Beaverton, OR, USA) and a two-terminal measurement chamber, respectively. Aluminum electrodes with a length of 4000 μm and spacing of 50 μm were defined through a shadow mask by thermal evaporator (Edwards Auto 306, Edwards Ltd., Burgess Hill, UK). The I versus V characteristics were obtained using a Keithley 4200-SCS measurement system (KEITHLEY, USA) Measurements were performed in the range of 0–200 V at a scan rate of 0.2 V/s, and the scan was repeated five times, consecutively (within 5 s). To investigate the environmental sensitivity, the device was kept in a vacuum (approximately 10^−1^ mbar) for over 2 days before starting each measurement.

## 3. Results

### 3.1. Vacuum Environment

The I versus V measurements were performed under a vacuum, repeated five times, and plotted as logI versus logV (Figure 1). The sample was placed under a vacuum for two days before conducting the measurements. A marginally higher current of 1.3 × 10^−6^ A is observed at the first measurement compared to that at the fifth measurement (8.7 × 10^−6^ A) at 200 V. Further, hysteresis is observed during the first measurement, which substantially reduces during the second measurement because the charge carriers with relatively lower release rates fill the traps as measurements are repeated. If the traps filled during one scan cannot release their charge before the subsequent measurement starts, the newly injected carriers easily flow without trapping. Further, the log⁡I versus log⁡V characteristics follow the $I\propto V^{m}$ relationship. A value of m is approximately 1 for <100 V in the first measurement, which changes to approximately 2 in the >100 V regimes. The data from the second to fifth measurements follow the power law $I\propto V^{2},$ suggesting that the dominant conduction mechanism in the first measurement is ohmic, which changes to space-charge-limited (SCL) conduction for the remaining scans [29,30,31,32,33].

Various factors affect conductivity, such as native defects in the film or surface contamination. However, surface contamination issues are negligible in a vacuum. The traps in the film could be attributed to the OH- groups, resulting from an incomplete dehydroxylation reaction owing to the low-temperature (<150 °C) processing conditions. In this study, we prepared a solution of zinc hydroxide in ammonia water for ZnO solution coating. Zinc hydroxide dissolves in ammonia water, causing a dehydration reaction and forming a water-soluble amine complex with hydroxyl ions. The dehydration reaction follows the following equation [34]:Zn(OH)_2_(s) + 4NH_3_(aq) → Zn(NH_3_)_4_^2+^(aq) + 2OH^−^(aq)

The OH^−^ groups still exist after low-temperature annealing treatment, which acts as electron trapping sites and deteriorates the device’s performance. This finding has been supported by previously published reports [35,36,37].

Figure 2 reveals log *I* versus log *V* plots for the solution-processed ZnO films after hydrogen plasma treatment in a vacuum. A 10 min plasma-treated ZnO film shows a large hysteresis evident in the first measurement, exhibiting a current of 6.0 × 10^−7^ A at 200 V, as shown in Figure 2a. This conductivity trend is similar to that observed for the untreated ZnO in vacuum (Figure 1). However, this behavior remarkably changes after 15 min of plasma treatment, exhibiting higher voltage and negligible hysteresis. A current peak of 1.1 × 10^−3^ A observed at 165 V reduces to 1.1 × 10^−4^ A at 200 V in the first measurement. Additionally, the value of m transforms from approximately 2 to 3 with an increase in the voltage from low to >100 V. The reaction between hydrogen radicals in the plasma and OH^−^ generated during the preparation of the solution-processed ZnO film results in oxygen vacancies, leading to higher conductivity and negligible hysteresis. These results are consistent with those reported previously [25]. The currents measured in vacuum are marginally higher than those measured in air or dry oxygen, with negligible hysteresis, which agrees with previously reported studies [38].

### 3.2. Air Environment

The dependence of the log⁡I versus log⁡V characteristics in the air environment as a function of the measurement sequence is shown in Figure 3. A current of 1.3 × 10^−5^ A is measured at 200 V during the first sweep, which increases to 2.4 × 10^−5^, 3.6 × 10^−5^, 4.6 × 10^−5^, and 8.2 × 10^−5^ A, with subsequent sweeps exhibiting a large hysteresis when the direction of the sweep is reversed. The current fit to $I\propto V^{m}$, with m≥3, irrespective of the measurement sequence. The oxygen absorption onto the film surface leads to lower conductivity due to the charge capture when the film is exposed to air. In contrast, these adsorbed molecules are eliminated in a vacuum environment, resulting in higher conductivity. However, the conductivity measured in air in this study is comparable to or even higher than that of other ambient air, in contrast to the previously reported results [9,10,11,12,13,14,15]. We infer that water reacts predominantly with solution-processed ZnO rather than oxygen from air. The solution-processed ZnO films used in this study are formed at a low temperature; therefore, the film contains a large quantity of unreacted OH^−^ ions, which readily react with water molecules on the surface [39]. This reaction leads to a reduced depletion region or the generation of extra free carriers [7,40,41], resulting in high conductivity.

Plots of log⁡I versus log⁡V in air for the solution-processed ZnO films as a function of the plasma treatment time is illustrated in Figure 4. A current of 2.7 × 10^−7^ A at 200 V following $I\propto V^{3}$ is observed for 10 min plasma-treated ZnO in the first measurement of the forward scan. In contrast, m decreases to approximately 2, with a large hysteresis between the forward and reverse scans. The current at 200 V decreases marginally with a constant value of m at approximately 2, as the measurement is repeated. Following plasma treatment for 15 and 30 min (Figure 4b,c), the sample exhibits currents of 1.0 × 10^−6^ and 2.3 × 10^−6^ A, respectively, at 200 V and the value of m is approximately 2. Negligible hysteresis is observed for 15 min plasma-treated ZnO, implying that the hydroxyl groups remaining in ZnO can be removed as H_2_O via the hydrogen plasma treatment. This leads to the elimination of water adsorption, resulting in low conductivity but good electrical stability compared to that of untreated ZnO [42,43]. Exposure to hydrogen plasma for 30 min exhibits large hysteresis resulting from excess hydrogen radicals acting as trap sites. Additionally, 15 min of plasma treatment is the optimal treatment time. The effect of the hydrogen plasma treatment on solution-processed ZnO films was demonstrated by an investigation of the electrical properties of ZnO TFTs in our previous study [25]. Solution-processed ZnO TFTs exhibited large hysteresis and Vth shifts due to the reaction between water molecules present in the environment and unreacted OH groups in the ZnO films. However, the device stability was enhanced after plasma treatment, exhibiting negligible hysteresis, a small Vth shift, and a low interface trap density (D_it_) value because the hydrogen plasma treatment accelerated the dihydroxylation/dehydration reactions, leading to a decrease and increase in carrier trapping and oxygen vacancies, respectively, resulting in TFTs with excellent electrical properties.

### 3.3. Oxygen Environment

Conductivity was measured in a dry oxygen environment to investigate the effect of oxygen. The samples were placed in a vacuum for two days before oxygen gas was added into the measurement chamber for 10 min at a flow rate of 50 cm^3^/min, and the I versus V characteristics were measured after waiting for 60 min. Figure 5 reveals a remarkable change in conductivity of the ZnO thin film in a dry oxygen environment compared to those measured in vacuum (Figure 1), with a value of approximately 3.9 × 10^−9^ A at 200 V during the first scan. Additionally, the value of m in $I\propto V^{m}$ is approximately 3, in contrast to that from Figure 1, wherein m=1 or 2.

The value of m≥3 suggests double-carrier space-charge-limited conductivity or space-charge-limited conductivity in the presence of trap states that are exponentially distributed, expressed by M. A. Lampert, Current Injection in Solids (New York: Academic) (1970), as given in Equation (1) [9].
(1)$$J=\left(\mu \frac{N_{c}}{q^{l-1}}\right){\left(\frac{2l+1}{l+1}\right)}^{\frac{1}{l}}{\left(\frac{\varepsilon_{0}k}{N_{t}}\cdot \frac{l}{l+1}\right)}^{l}\cdot \frac{V^{l+1}}{d^{2l+1}}$$
where l is an exponent having a value greater than 1 [9].

The current, especially for the first measurement, reveals a remarkable oscillatory behavior, as illustrated in Figure 5 (inset). The oscillation period and current values increase with increasing applied voltage and multiple measurement sweeps. The oxygen is adsorbed on the ZnO surface as negatively charged ions, decreasing the electrical conductivity [9,10,11,12,13,14,15]. Further, the sensitivity to the measurement environment highly depends on the film thickness. The thick film forms a conductive path in the bulk region and exhibits a large and dense grain structure. Additionally, even surfaces, compared with thin films, produce long electron mean free paths, undisturbed by absorbed molecules such as negatively charged oxygen ions [44,45]. Therefore, the dominant factors determining the electrical behavior of thick films are their bulk properties, including native defects, with negligible environmental effects. In contrast, the thin film exhibits a rough surface, and the conductive path is extremely close to surface contamination. Each trough in the roughness acts as a trapping site, thus inducing a surface space charge region. Therefore, the charge trapped by the roughness results in a high barrier, which disturbs the conduction path (see the inset in Figure 6a). These issues are schematically illustrated in Figure 6.

The oscillation behavior observed in Figure 5 could be attributed to carrier accumulation and dispersal. Injected carriers accumulate at the grain boundaries as they have insufficient energy to overcome the high barrier generated by the trapped charge, as shown schematically in Figure 6a (inset). However, these carriers overcome the barrier at a specific applied electric field, rapidly increasing the current value followed by a sharp decrease resulting from carrier dispersal. Repeated carrier accumulation, followed by their dispersal, leads to oscillations in the I versus V characteristics. These findings could be further corroborated by determining the current versus voltage characteristics of films with different thicknesses and over a range of temperatures in future research.

Figure 7 illustrates the change in the log⁡I versus log⁡V characteristics for a solution-processed ZnO film after hydrogen plasma treatment at various times in dry oxygen. The first scan in Figure 7a reveals a current of 6.0 ×10^−8^ A at a bias of 200 V, with m value of approximately 3, indicating similar conductivity properties to those of the untreated films in an oxygen environment (Figure 5). The conductivity increases by more than one order of magnitude with increasing plasma treatment time, as shown in Figure 7b,c. The oscillatory behavior observed in Figure 5 is absent. Additionally, the conductivity is comparable to that measured in air or vacuum environments after plasma treatment, indicating that the films are unaffected by oxygen molecules, confirming that plasma treatment increases the conductivity and induces a passivation effect.

### 3.4. Electrical Transport Characteristics

To understand the change in the electrical transport characteristics, the current versus voltage characteristics of a solution-processed ZnO film were measured as a function of temperature within the 300–350 K range at a bias scan rate of 0.4 V/s. A current of 8.1 × 10^−9^ A is observed at 20 V and 300 K, lower than that at 350 K (4.4 × 10^−7^ A), indicating that the solution-processed ZnO films exhibit semiconducting behavior.

Figure 8a reveals a considerable hysteresis in the *I*–*V* characteristics, expected to originate from the trapping and de-trapping of charge carriers, as mentioned previously. The counterclockwise nature of the hysteresis indicates that the trapping rate is faster than the de-trapping rate, leading to a higher current in the reverse scan. Figure 8b shows I versus V curves at 350 K and different voltage scan rates, achieved by varying the delay between voltage steps from 0.1 to 60 s. A substantial dependence of hysteresis on the delay time is observed. A longer delay time produces negligible hysteresis, indicating that 60 s is sufficient for releasing the trapped charges.

Figure 9 shows the plot of I versus T for the solution-processed ZnO films before and after plasma treatment for 15 min. The applied voltages are fixed at 0.5 (Figure 9a) and 20 V (Figure 9b). Both curves increase exponentially with increasing temperature, indicating semiconducting behavior.

Figure 9 is replotted in $\ln I\;versus\;T^{-1}$ form and the plots are illustrated in Figure 10. The activation energies of the solution-processed ZnO films before and after plasma treatment were calculated from the slope of the linear fit according to Equation (2) [46,47]:(2)$$I=I_{0}\exp(-\frac{E_{a}}{k_{B}T})$$

The origin of a deep donor level (activation energy of 0.7–0.8 eV approximately) is ascribed to native point defects such as oxygen vacancies, whereas an activation energy of 0.3–0.6 eV could be associated with a shallow donor level [48,49]. The activation energies calculated for both samples are listed in Table 1. The activation energy of approximately 0.6 eV for untreated ZnO film originates from a particular trap level. In contrast, the ZnO film plasma-treated for 15 min shows a relatively high activation energy of 0.84 eV at 20 V, suggesting that an extra oxygen vacancy generated by the reaction of unreacted OH^−^ and hydrogen radicals in the plasma dominantly acts as a larger number of other traps at 20 V. However, an in-depth study is required to facilitate a better understanding of the origin of the rapidly increasing current in the high-voltage region (>100 V), as shown in Figure 2b.

## 4. Conclusions

The effects of different measurement conditions on the electrical conductivity of solution-processed ZnO films were investigated. A dry oxygen environment resulted in degradation of the conductivity owing to the absorption of negatively charged oxygen molecules. The conductivity of the solution-processed ZnO films measured in air was comparable to that measured in vacuum, inferring that the water molecules in air predominantly reacted with the OH^−^ groups in the films, resulting in a reduced depletion region or the generation of extra free carriers. Hydrogen plasma removes the remaining OH^−^ in the form of H_2_O after low-temperature processing and stabilizes the environmental sensitivity of the solution-processed ZnO film. Additionally, hydrogen acts as a shallow donor, facilitating the creation of additional oxygen vacancies, which improves conductivity. The quantity of hydrogen was controlled by the plasma treatment time, and optimal time for ZnO film treatment was 15 min. This study focused on the effects of OH traps generated in ZnO thin films via a low-temperature solution process in various environments and the electrical conductivity mechanism of films using hydrogen plasma treatment. These research findings could be helpful to better understand the stability and reliability of devices in various environments and enhance their potential applications.

## Figures and Tables

**Figure 1 materials-17-02673-f001:**
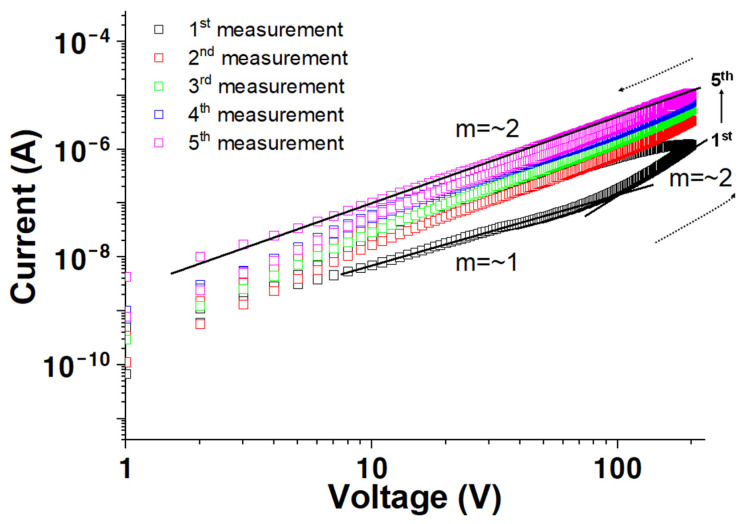
Log *I* versus log *V* measured in vacuum as a function of measurement sequence for a solution-processed ZnO film.

**Figure 2 materials-17-02673-f002:**
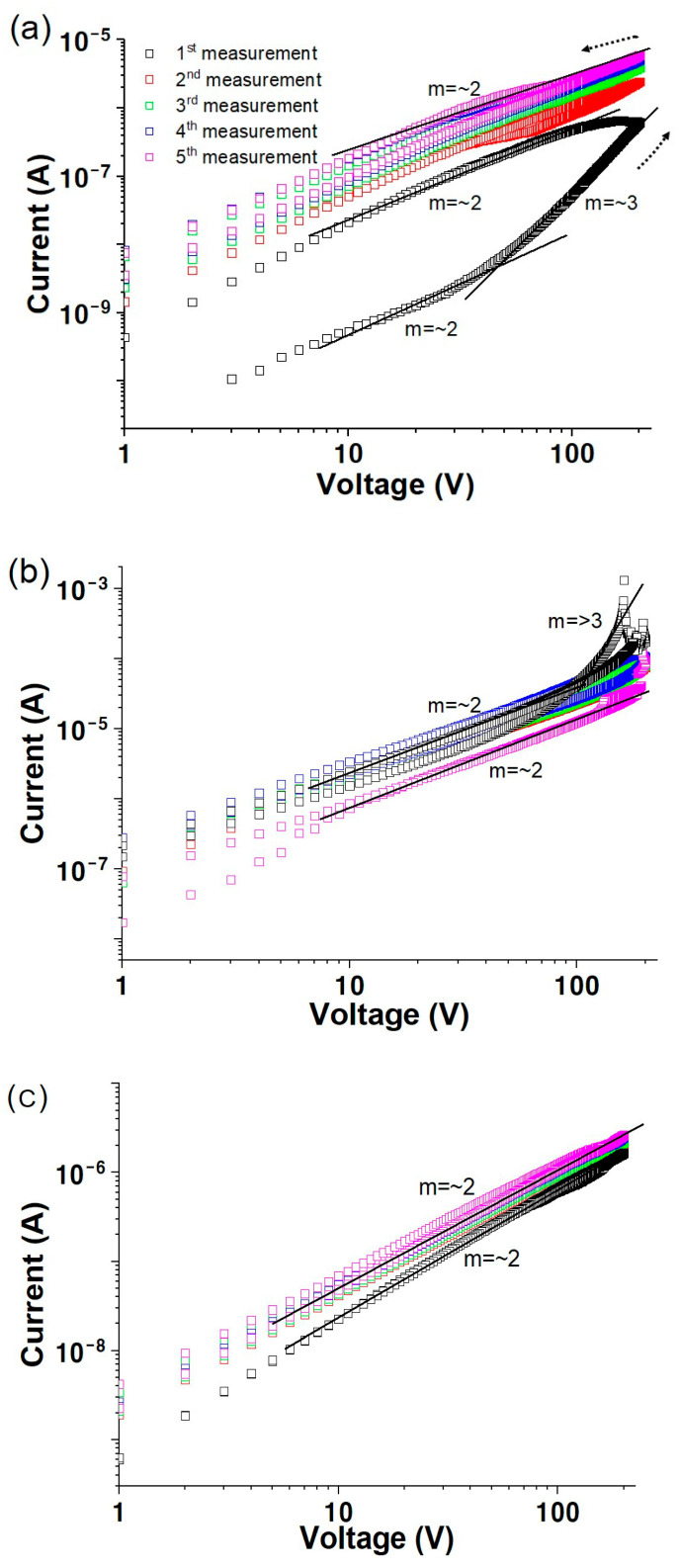
Change in conductivity of solution-processed ZnO with hydrogen plasma treatment times for (**a**) 10 min, (**b**) 15 min, and (**c**) 30 min as a function of measurement sequence in vacuum environment.

**Figure 3 materials-17-02673-f003:**
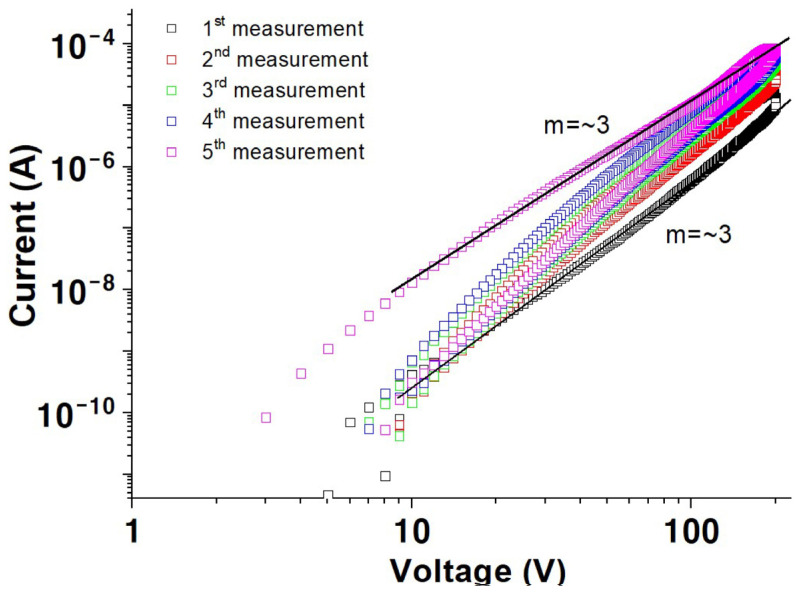
Log *I* versus log *V* measured in air on the number of measurements sweeps for a solution-processed ZnO film.

**Figure 4 materials-17-02673-f004:**
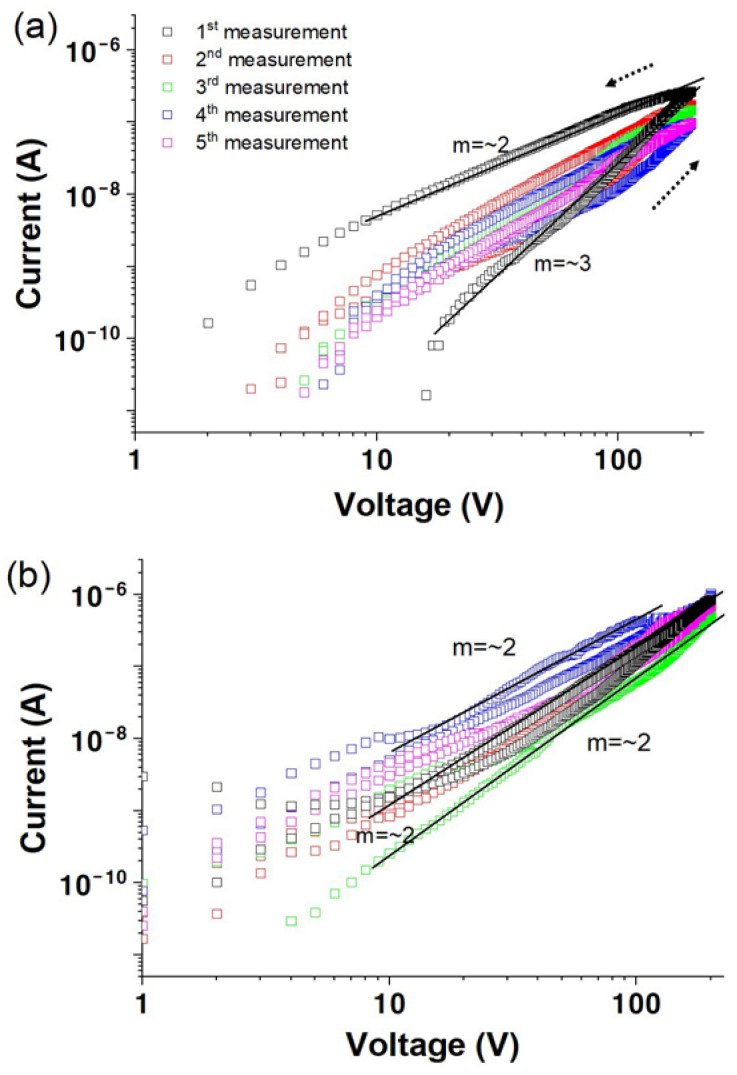
Log *I* versus log *V* for solution-processed ZnO films after hydrogen plasma treatment for (**a**) 10 min, (**b**) 15 min, and (**c**) 30 min as a function of measurement sequence in air.

**Figure 5 materials-17-02673-f005:**
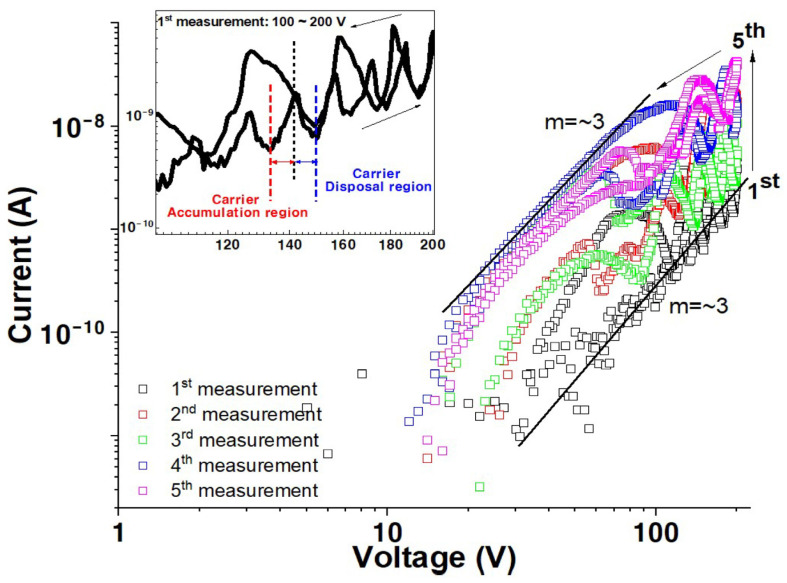
Log *I* versus log *V* measured in oxygen as a function of measurement sequence for a solution-processed ZnO film.

**Figure 6 materials-17-02673-f006:**
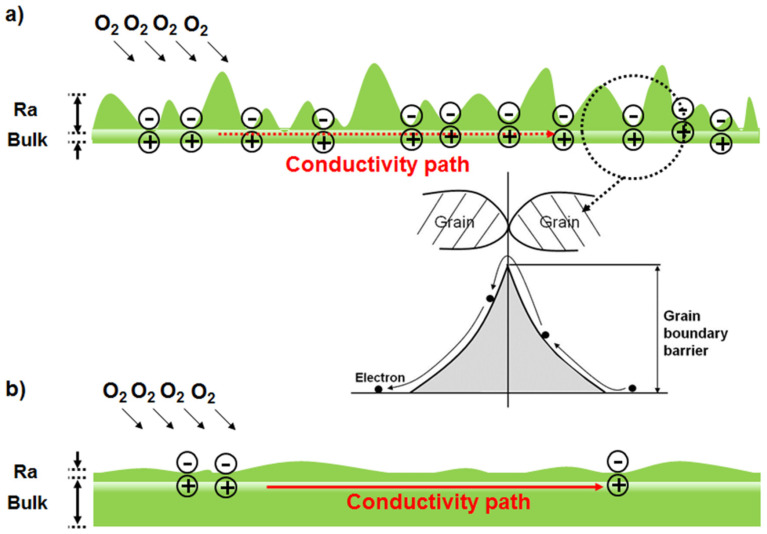
Schematic diagrams of the conductive path through (**a**) a thin film and (**b**) a thick film.

**Figure 7 materials-17-02673-f007:**
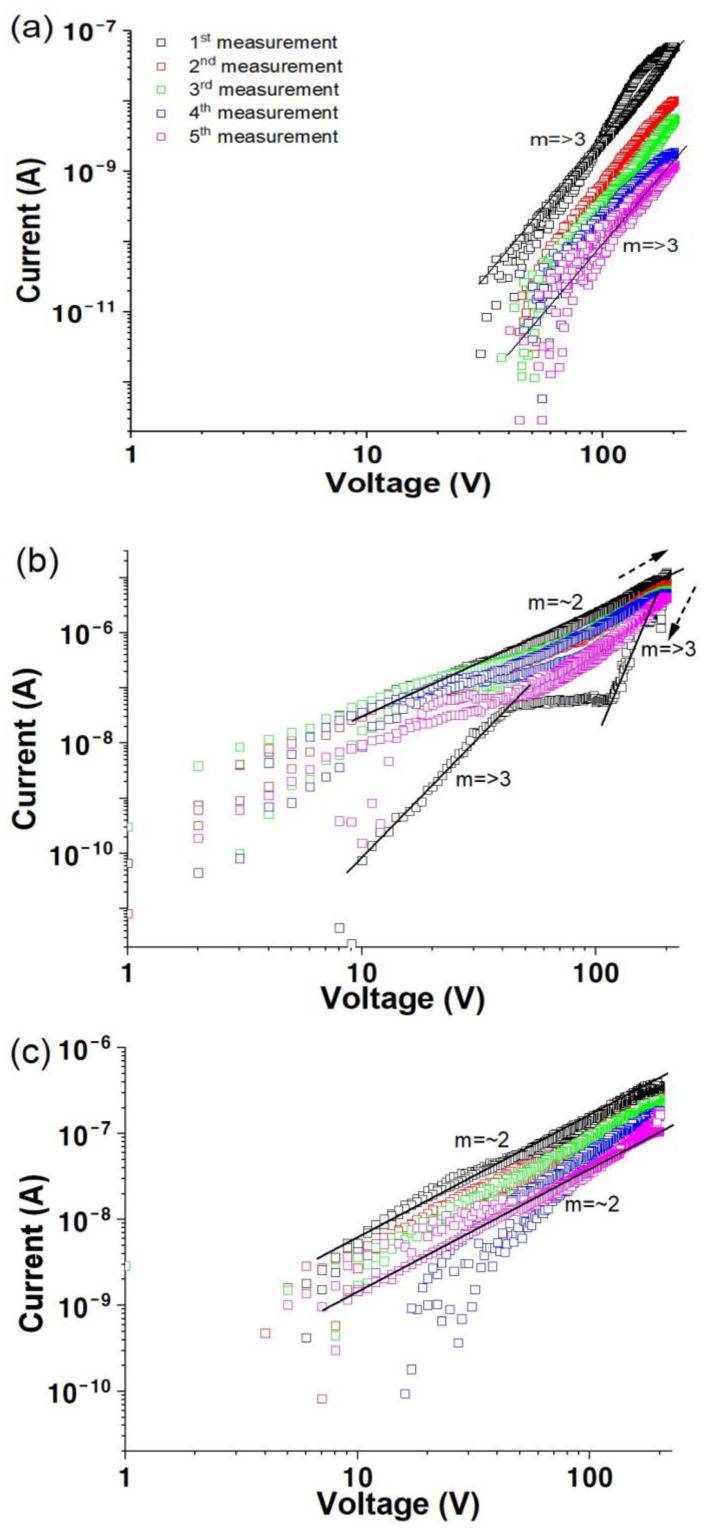
Log *I* versus log *V* for solution-processed ZnO films after hydrogen plasma treatment for (**a**) 10 min, (**b**) 15 min, and (**c**) 30 min as a function of measurement sequence in dry oxygen.

**Figure 8 materials-17-02673-f008:**
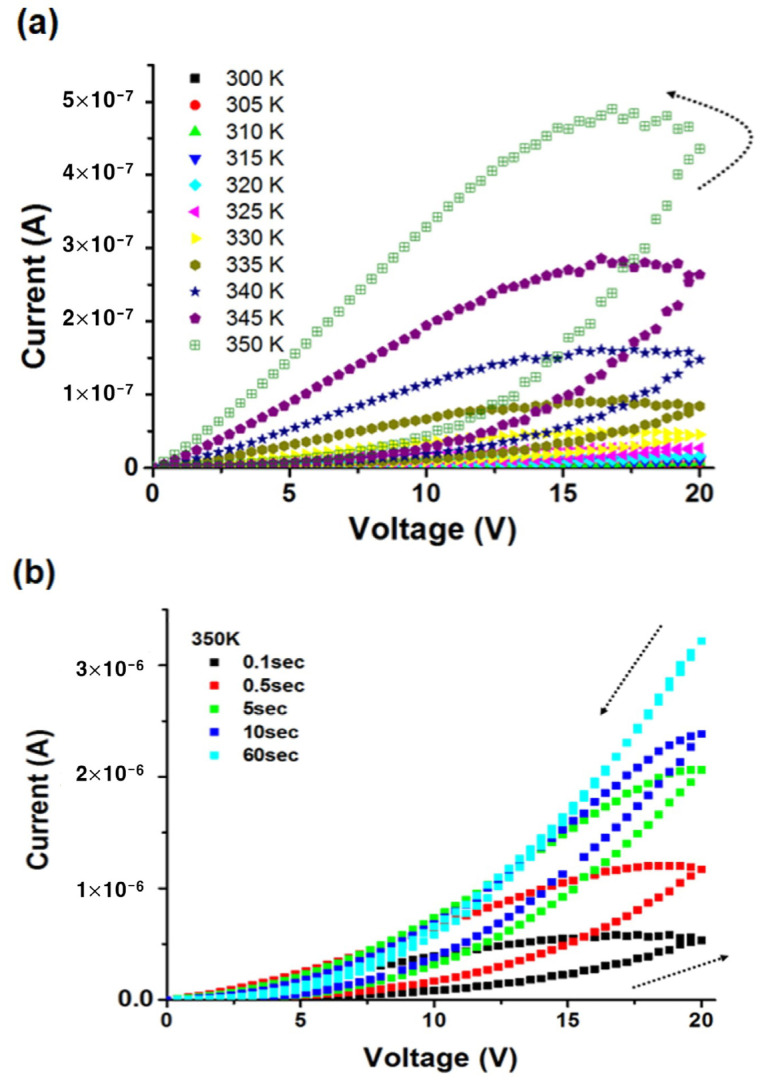
(**a**) *I* versus *V* characteristics of a solution-processed ZnO film in helium ambient as a function of (**a**) temperature between 300 K and 350 K, (**b**) measurement delay time at a temperature of 350 K.

**Figure 9 materials-17-02673-f009:**
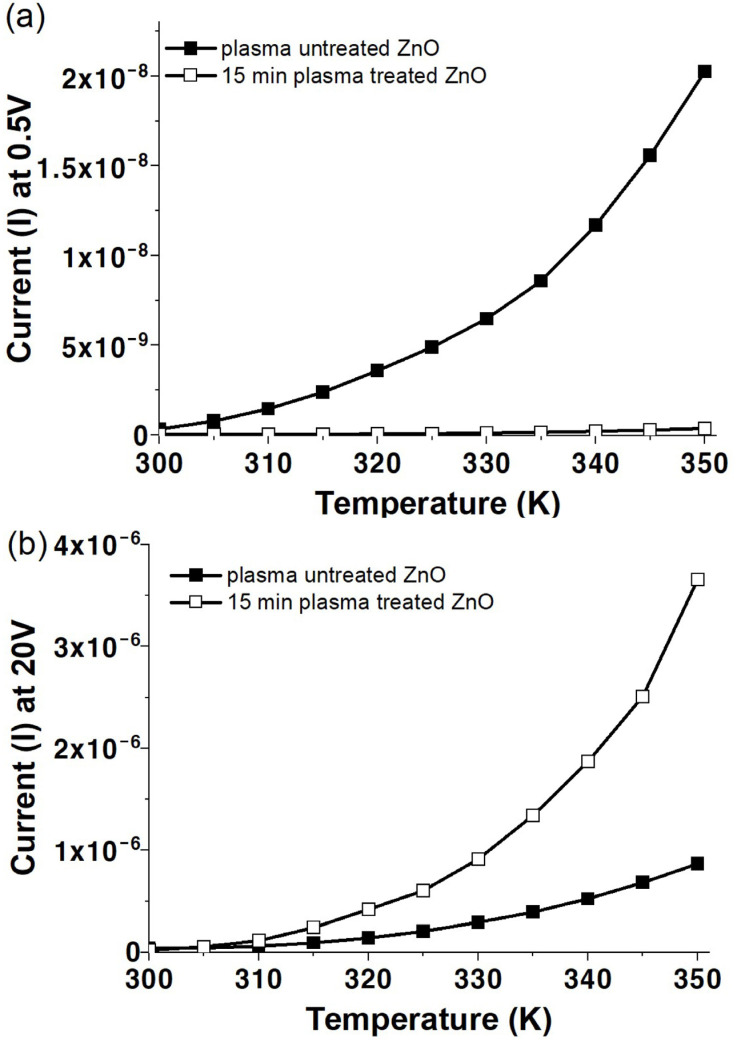
Current versus temperature properties for ZnO films before and after 15 min of plasma treatment, measured at applied voltages of (**a**) 0.5 V and (**b**) 20 V.

**Figure 10 materials-17-02673-f010:**
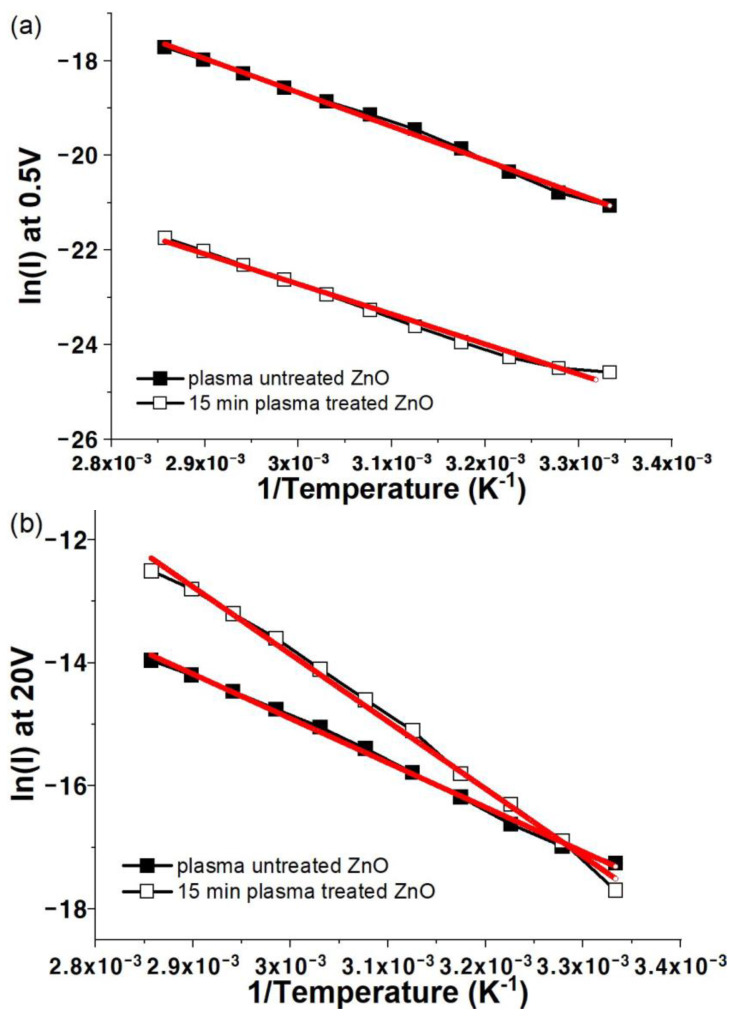
Ln *I* versus 1/*T* characteristics for ZnO films before and after 15 min of plasma treatment, measured at applied voltages of (**a**) 0.5 V and (**b**) 20 V.

**Table 1 materials-17-02673-t001:** Activation energy summary of solution-processed ZnO films before and after oxygen plasma treatment.

	0.5 V	20 V
Solution-processed ZnO	0.64 ± 0.01	0.67 ± 0.01
15 min plasma-treated ZnO	0.71 ± 0.10	0.84 ± 0.20

## Data Availability

Data are contained within the article.

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
