# Peer review of "Effects of Hydrogen Plasma Treatment on the Electrical Behavior of Solution-Processed ZnO Thin Films"

_materials, 2024, doi:10.3390/ma17112673_

Round 1

Reviewer 1 Report

Comments and Suggestions for Authors

Comments and Suggestions for Authors

In this work the authors studied the effects of atmospheric hydrogen plasma treatment on the in-plane conductivity of solution-processed zinc oxide (ZnO) under different environments such as vacuum, air and oxygen with and without hydrogen-plasma treatments. Hydrogen plasma treatment resulted in stabilization of the environmental sensitivity of these solution processed ZnO films. 

This is an extensive work on the electrical behavior of solution processed ZnO thin films. However, the manuscript needs to be revised and the following main points need to be carefully addressed before it can be considered for publication: 

1.     First, the added value of this review paper among the large literature in the field should be clearly addressed in the manuscript. 

2.     The introduction needs to be improved. The authors write that: “For most of these devices, the material is required in the form of a thin film, which is very sensitive to ambient conditions [1]. This absorbed molecule from the environment, especially oxygen molecules, induce a decrease in the electrical properties due to the surface energy band bending [2].” Which absorbed molecule is it?

This sentence should be clarified/modified. 

3.     The references [4], [5], and [6] appear inappropriate because they do not regard ZnO materials/structures, dealing instead with Silicon, silicon-carbide and GaN, respectively, as results even from their title: [4] Marsh, O.J.; Wiswanathan, C.R. Space-Charge Limited Current of Holes in Silicon and Techniques for Distinguishing Double and Single Injection. J. Appl. Phys. 1967, 38, 3135. [5] Tehrani, S.; Kim, J.S.; Hench, L.L.; Van Vliet, C.M.; Bosman, G. Observation of single-carrier‚ space-charge-limited flow in nitrogen doped alfa-silicon carbide. I. I-V characteristics and impedance. J. Appl. Phys. 1985, 58, 1562. [6] Hall, H.P.; Awaah, M.A.; Das, K. Deep-level dominated rectifying contacts for n-type GaN films. Phys. Stat. Sol. (a). 2004, 201, 320 522-528. They should, therefore, be replaced or at least integrated with more appropriate references specifically related to the ZnO material. 

4.     Still in the introduction, to give to the readers a complete overview on the material, it is required to briefly mention some complementary synthesis technologies used in the literature for the fabrication of ZnO-based structures, including at least the carbo-thermal transport growth [DOI:10.1007/s00339-007-3946-4] and electron beam evaporation [https://doi.org/10.1016/j.apsusc.2008.09.026]. 

5.     In the section “2. Materials and Methods” the brand of the precursors used in solution and the one of the electric measurement instrumentations are missing. Also, the parameters of the grown films (thickness, morphological and structural quality and so on) of the specific samples studied in the present paper, should be reported. 

6.     At lines 94-96, the statement regarding the origin of the traps related to the dehydroxylation reactions should be clarified by including the relative chemical reaction. 

7.     At lines 108-110 the statements should be supported by a reference. 

8.     At lines 180-181: along with the name of the author Lampert, the reference [3] should be reported, however at line 183 Ref [34], regarding organic crystals is instead reported. This point should be clarified/amended with the appropriate and relevant reference. 

9.  Fig. 5 is not clear enough and needs to be better represented. 

10.  Fig. 9 (a) and (b): what the lines represent? Are they just a line for the eye or the result of a fit? 

11.  Fig. 10: the values of the x-axis should be reported in the scientific notation. 

12.  The expression “un-plasma treated ZnO” reported in Figs 9 and 10 should be replaced with “plasma untreated ZnO”. 

13.  in Table 1, the errors reported for the values of “15 min plasma treated ZnO” should have the same number of decimal places of the relative values, i.e. two. 

14.  Furthermore, the conclusions should highlight the value that the manuscript adds to the current literature in the field and outline some perspectives. 

Reviewer 2 Report

Comments and Suggestions for Authors

The manuscript titled "Effects of Hydrogen Plasma Treatment on the Electrical Behaviour of Solution-processed ZnO Thin Films" presents an interesting study on the influence of measurement environment and hydrogen plasma treatment on the in-plane electrical conductivity of solution-processed zinc oxide films. The topic is relevant and the findings could be useful for optimizing the performance and stability of ZnO-based electronic devices. However, there are several aspects that need to be clarified and elaborated to enhance the quality and impact of the paper:

 1.       The introduction provides a good background on the environmental sensitivity of solution-processed ZnO films and the potential benefits of hydrogen doping. However, the novelty and specific objectives of this work could be stated more explicitly. How does your study advance the current understanding of the conduction mechanisms and stability of ZnO films?

2.       The experimental section describes the fabrication and characterization methods, but some additional details would be helpful. For example, what is the thickness of the ZnO films? How was the film thickness measured? The electrical properties could depend strongly on the film thickness, so this information is important for interpreting the results.

3.       In Section 3.1, the authors attribute the increase in current with repeated voltage sweeps in vacuum to trap filling effects. This is a reasonable explanation, but it would be more convincing if supported by additional evidence or discussion. For example, what is the nature and origin of these traps? How does the trap density compare with other reports on solution-processed ZnO?

4.       The observation of oscillatory current behavior in oxygen environment (Section 3.3) is intriguing, but the proposed mechanism of carrier accumulation and dispersal at grain boundaries needs further justification. Have similar oscillations been reported in other studies of ZnO or related materials? It would be valuable to provide more insights into the dynamics and timescales of these processes.

5.       The activation energy analysis in Section 3.4 provides useful information on the trap levels in untreated and plasma-treated ZnO films. However, the physical origin and implications of these trap levels merit further discussion. Are they related to specific defects or impurities? How do they compare with the literature values for ZnO prepared by other methods?

6.       The conclusion section summarizes the key findings, but could be strengthened by providing clearer insights into the underlying mechanisms and potential applications. For example, what is the role of hydrogen in passivating surface states and enhancing stability? How do the present results guide the optimization of plasma treatment conditions for practical devices?

7.       The overall structure and presentation of the paper are good, but there are some grammatical errors and awkward phrases that need to be polished. For example, in the abstract, "This study reports on the effects of atmospheric hydrogen plasma treatment on the in-plane conductivity of solution-processed zinc oxide (ZnO) under various environments" could be rephrased as "This study investigates the effects of atmospheric hydrogen plasma treatment on the in-plane conductivity of solution-processed zinc oxide (ZnO) films under various environments".

Comments on the Quality of English Language

Moderate editing of English language required

Round 2

Reviewer 1 Report

Comments and Suggestions for Authors

The authors have correctly addressed all the issues and now the manuscript is ready for publication in the journal. 

Reviewer 2 Report

Comments and Suggestions for Authors

The current version of the manuscript is acceptable for publication without further revisions.